# Joint Modeling of Visual Objects and Relations for Scene Graph Generation

**Minghao Xu**[1]    **Meng Qu**[2,3]    **Bingbing Ni**[1]    **Jian Tang**[2,4,5]
[1]Shanghai Jiao Tong University, Shanghai 200240, China    [2]Mila - Québec AI Institute
[3]University of Montréal    [4]HEC Montréal    [5]CIFAR AI Research Chair
{xuminghao118, nibingbing}@sjtu.edu.cn    meng.qu@umontreal.ca    jian.tang@hec.ca

## Abstract

An in-depth scene understanding usually requires recognizing all the objects and their relations in an image, encoded as a scene graph. Most existing approaches for scene graph generation first independently recognize each object and then predict their relations independently. Though these approaches are very efficient, they ignore the dependency between different objects as well as between their relations. In this paper, we propose a principled approach to jointly predict the entire scene graph by fully capturing the dependency between different objects and between their relations. Specifically, we establish a unified conditional random field (CRF) to model the joint distribution of all the objects and their relations in a scene graph. We carefully design the potential functions to enable relational reasoning among different objects according to knowledge graph embedding methods. We further propose an efficient and effective algorithm for inference based on mean-field variational inference, in which we first provide a warm initialization by independently predicting the objects and their relations according to the current model, followed by a few iterations of relational reasoning. Experimental results on both the relationship retrieval and zero-shot relationship retrieval tasks prove the efficiency and efficacy of our proposed approach.

## 1   Introduction

Modern object recognition [32, 10, 35] and detection [28, 27, 57] systems excel at the perception of visual objects, which has significantly boosted many industrial applications such as intelligent surveillance [18, 49] and autonomous driving [23, 38]. To have a deeper understanding of a visual scene, detecting and recognizing the objects in the scene is however insufficient. Instead, a comprehensive cognition of visual objects and their relationships is more desirable. Scene Graph Generation (SGG) [13] is a natural way to achieve this goal, in which a graph incorporating all objects and their relations within a scene image is derived to represent its semantic structure.

Most previous works for SGG [48, 55, 53, 36, 4, 37] usually first independently predict different objects in a scene and then predict their relations independently. In practice, though such methods are very efficient, they ignore the dependency between different objects and between the relations of different object pairs. For example, a `car` could frequently co-occur with a `street`, and the relation `eating` could always appear along with the relation `sitting on`. Modeling such dependency could be very important for accurate scene graph prediction, especially for rare objects and relations. There are indeed some recent works [6, 5] along this direction. For example, Dai et al. [6] explored the triplet-level label dependency among a head object, a tail object and their relation. These methods have shown very promising results, while they only explored the limited dependency within a triplet. How to capture the full dependency between different objects and between their relations within a whole scene graph remains very challenging and unexplored.

35th Conference on Neural Information Processing Systems (NeurIPS 2021).

To attain such a goal, in this paper, we propose a principled approach called **J**oint **M**odeling for **S**cene **G**raph **G**eneration (*JM-SGG*) to predict the whole scene graph by jointly capturing all the label dependency within it, *i.e.* the dependency between different objects and their relations and also the interdependency between them. Specifically, we model the joint distribution of all objects and relations in a scene graph with the conditional random field (CRF) framework [17]. To flexibly model the joint distribution, the key is to define effective potential functions on both nodes (*i.e.* objects) and edges (*i.e.* relations between objects). We define the potential functions on objects according to the object representations extracted by existing neural network based object detector. It is however nontrivial to design effective potential functions on edges, since these potential functions have to capture the relation between two objects in an edge and meanwhile allow relational reasoning among different edges, which models the dependency among the relations on various edges. Inspired by the existing work of knowledge graph embedding [20], which represents entities and relations in the same embedding space and performs relational reasoning in that space, we define our potential functions according to the knowledge graph embedding method and hence allow efficient relational reasoning between different object pairs in a scene graph.

Such a fully expressive model also brings challenges to both learning and inference due to the complicated structures between different random variables in the CRF, *i.e.* objects and their relations. We therefore further propose an efficient and effective inference algorithm based on mean-field variational inference, which is able to assist the gradient estimation for learning and derive the most likely scene graph for test. Traditional mean-field methods usually suffer from the problem of slow convergence. Instead of starting from a randomly initialized variational distribution as in traditional mean-field methods, we propose to initialize the variational distribution, *i.e.* the marginal distribution of each object and each relation, with a factorized tweak of JM-SGG model, and then perform a few iterations of message passing induced by the fixed-point optimality condition of mean field to refine the variational distribution, which allows our approach to enjoy both good precision and efficiency.

To summarize, in this paper, we make the following contributions:

- We propose Joint Modeling for Scene Graph Generation (JM-SGG) which is a fully expressive model that can capture all the label dependency in a whole scene graph.
- We propose a principled mean-field variational inference algorithm to enable the efficient learning and inference of JM-SGG model.
- We verify the superior performance of our method on both relationship retrieval and zero-shot relationship retrieval tasks under various settings and metrics. Also, we illustrate the efficiency and efficacy of the proposed inference algorithm by thorough analytical experiments.

## 2    Related Work

**Scene Graph Generation (SGG).** This task aims to extract structured representations from scene images [13], including the category of objects and their relationships. Previous works performed SGG by propagating the information from different local regions [48, 53, 50, 36], introducing external knowledge [9, 52], employing well-designed loss functions [56, 14, 34] and performing unbiased scene graph prediction [4, 19, 37]. Most of these methods predict each object and relation label independently based on an informative representation, which fails to capture the rich label dependency within a scene graph and is thus less expressive. Several former works [6, 5] attempted to model such label dependency within a single relational triplet but not on the whole scene graph.

*Improvements over existing methods.* The proposed JM-SGG model is, to our best knowledge, the first approach that jointly models all the label dependency within a scene graph, including the one within object or relation labels and the one between these two kinds of labels. To attain this goal, a unified CRF is constructed for graphical modeling, and a mean-field variational inference algorithm is designed for efficient learning and inference, which show technical contributions.

**Conditional Random Fields (CRFs).** CRFs are a class of probabilistic graphical modeling methods which perform structured prediction upon the observed data. CRF-based approaches have been broadly studied on various computer vision problems, including segmentation [17, 43, 51, 26], super-resolution [39, 46], image denoising [29, 42] and scene graph generation [6, 5]. These former works utilizing CRF for SGG [6, 5] aimed to model the conditional distribution of a single triplet upon visual representations. By comparison, our approach models the conditional distribution of a whole scene graph upon the observed scene image, which is more expressive.

# 3 Problem Definition and Preliminary

## 3.1 Problem Definition

This work focuses on extracting a *scene graph*, *i.e.* a structured representation of visual scene [13], from an image. Formally, we define a scene graph as $G = (y_O, R)$. $y_O$ denotes the category labels of all objects $O$ in the image, and it holds that $y_o \in \mathcal{C}$ for each object $o \in O$, where $\mathcal{C}$ stands for the set of all object categories, including the "background" category. $R = \{(o_h, r, o_t)\}$ is the set of relational triplets/edges with $r \in \mathcal{T}$ as the relation type from head object $o_h$ to tail object $o_t$ ($o_h, o_t \in O$), where $\mathcal{T}$ represents all relation types, including the type of "no relation". In this work, we aim at *jointly* modeling visual objects and visual relations as defined below:

**Joint Scene Graph Modeling.** Given an image $I$, we aim to jointly predict object categories $y_O$ and the relationships $R$ among all objects, which models the joint distribution of scene graphs, *i.e.* $p(G|I) = p(y_O, R|I)$, with comprehensively considering the dependency within $y_O$ and $R$ and also the interdependency between them.

## 3.2 Conditional Random Fields

Conditional Random Field (CRF) is a discriminative undirected graphical model. Given a set of observed variables $\mathbf{x}$, it models the joint distribution of labels $\mathbf{y}$ based on a Markov network $\mathcal{G}$ that specifies the dependency among all variables:

$$p(\mathbf{y}|\mathbf{x}) = \frac{1}{Z(\mathbf{x})} \prod_C \phi_C(\mathbf{x}_C, \mathbf{y}_C), \quad Z(\mathbf{x}) = \sum_{\mathbf{y}} \prod_C \phi_C(\mathbf{x}_C, \mathbf{y}_C). \tag{1}$$

where $\phi_C$ denotes the nonnegative potential function defined over the variables in clique $C$ (a clique is a fully-connected local subgraph), and $Z(\mathbf{x})$ is a normalization constant called partition function.

# 4 Model

In this section, we introduce Joint Modeling for Scene Graph Generation (*JM-SGG*). Current methods solve the problem by independently predicting each object and relation label upon an informative representation, and thus the prediction of different labels cannot fully benefit each other. JM-SGG tackles the limitation by jointly modeling all the objects and relationships in a visual scene with a unified conditional random field, which enables the prediction of various object and relation labels to sufficiently interact with each other. Nevertheless, learning and inferring this complex CRF is nontrivial, and we thus propose to use maximum likelihood estimation combined with mean-field variational inference, yielding an efficient algorithm for learning and inference. Next, we elucidate the details of our approach.

## 4.1 Representation

In the JM-SGG model, we organize the observed scene image $I$ and all object and relation labels in the latent scene graph (*i.e.* $y_O$ and $R$) as the nodes in a unified conditional random field. Since the interactions of these

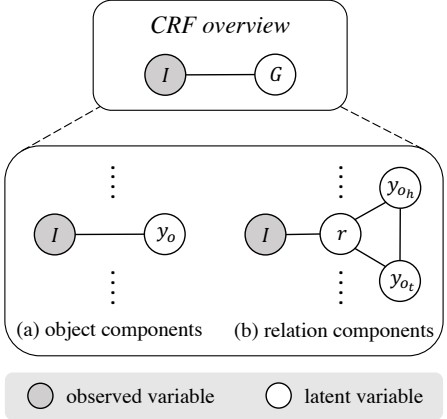

Figure 1: The CRF model for scene graph. (a) An *object component* models the dependency of object category $y_o$ on image $I$. (b) A *relation component* models the dependency of relation type $r$ on image $I$ and also the *interdependency* among object and relation labels (*i.e.* $y_{o_h}$, $y_{o_t}$ and $r$).

nodes are either for a single object or for the relationship between an object pair, we decompose the graphical structure of whole network into two sets of components. (1) *Object components*: For an object $o \in O$, we consider the dependency of its category label on its visual representation and thus connect $y_o$ with $I$, as shown in Fig. 1(a). (2) *Relation components*: for a relational triplet $(o_h, r, o_t) \in R$, we consider the dependency of relation type $r$ on the visual cues in image $I$, and we also model the *interdependency* among the object and relation labels in this triplet (*i.e.* $y_{o_h}$, $y_{o_t}$

and $r$), which forms a relation component as Fig. 1(b) shows. By combining all object and relation components, the CRF can capture the comprehensive label dependency within a scene graph. We now define the joint distribution of scene graphs upon the observed scene image as below:

$$p_{\Theta}(G|I) = \frac{1}{Z_{\Theta}(I)} f_{\Theta}(G, I), \tag{2}$$

$$f_{\Theta}(G, I) = \prod_{o \in O} \phi(y_o, I) \prod_{(o_h, r, o_t) \in R} \psi(r, y_{o_h}, y_{o_t}, I), \tag{3}$$

where $\Theta$ summarizes the parameters of whole model, $f_{\Theta}$ is an unnormalized likelihood function, $Z_{\Theta}$ denotes the partition function, and $\phi$ and $\psi$ are the potential functions defined on object and relation components, respectively. Next, we define these potential functions based on the extracted visual representations and the correlation among different labels.

**Visual representation extraction.** Given a scene image $I$, we first utilize a standard object detector (*e.g.* Faster R-CNN [28] in our implementation) to obtain a set of bounding boxes which potentially contain the objects in the image, and object representations $z_O = \{z_o | o \in O\}$ ($z_o \in \mathbb{R}^D$) are then derived by RoIAlign [11]. We regard the union bounding box over a pair of objects as their context region and again use RoIAlign to get all context representations $z_R = \{z_{ht} | (o_h, r, o_t) \in R\}$ ($z_{ht} \in \mathbb{R}^D$). Here, $D$ denotes the latent dimension of objects and contexts. By denoting the whole object detector as $g_\theta$, this feature extraction process can be represented as: $(z_O, z_R) = g_\theta(I)$.

**Potential function definition.** The potential function $\phi(y_o, I)$ for object component models the dependency of object category $y_o$ on object representation $z_o$ by measuring their affinity. To conduct such a measure, we represent each object category with a prototype [33] (*i.e.* a learnable embedding vector) in the continuous space, which forms a prototype set $\mathbf{C} = \{\mathbf{C}_i \in \mathbb{R}^D | i \in \mathcal{C}\}$ for all object categories ($D$ denotes the dimension of object space). On such basis, we define $\phi(y_o, I)$ by computing the distance between object representation $z_o$ and the prototype of object category $y_o$:

$$\phi(y_o, I) = \exp\left(-d(\mathbf{C}_{y_o}, z_o)\right), \tag{4}$$

where $d$ is a distance measure (*e.g.* Euclidean distance in our practice).

The potential function $\psi(r, y_{o_h}, y_{o_t}, I)$ for relation component models the dependency of relation type $r$ on the relevant visual representations in image $I$, and it also models the interdependency among the object and relation labels of a triplet (*i.e.* $y_{o_h}$, $y_{o_t}$ and $r$). Therefore, we can factorize $\psi(r, y_{o_h}, y_{o_t}, I)$ into a term $\psi_{\text{visual}}(r, I)$ for modeling visual influence and another term $\psi_{\text{triplet}}(r, y_{o_h}, y_{o_t})$ for modeling the label consistency within a triplet:

$$\psi(r, y_{o_h}, y_{o_t}, I) = \psi_{\text{visual}}(r, I) \, \psi_{\text{triplet}}(r, y_{o_h}, y_{o_t}). \tag{5}$$

Similarly, for measuring $r$ in the continuous space, a prototype set $\mathbf{T} = \{\mathbf{T}_j \in \mathbb{R}^K | j \in \mathcal{T}\}$ is constructed for all relation types ($K$ denotes the dimension of relation space).

We consider two kinds of visual representations that affect the prediction of relation type $r$, *i.e.* the context representation $z_{ht}$ and the head and tail object representations $z_{o_h}$ and $z_{o_t}$. The influence of context representation can be easily measured by projecting context representation $z_{ht}$ to the relation space and computing its distance to the prototype of relation type $r$. However, measuring the influence of head and tail object representations and evaluating the label consistency within a triplet are nontrivial, which require to model the ternary correlation among head object, tail object and their relationship. Inspired by the idea of TransR [20], an effective knowledge graph embedding technique, we model such ternary correlation by treating each relation as a translation vector from head object embedding to tail object embedding in the same embedding space. Specifically, we first apply the translation vector $\mathbf{T}_r$ specified by relation $r$ to head object embedding, and then compute the distance between the translated embedding and tail object embedding. Based on these thoughts, we define $\psi_{\text{visual}}(r, I)$ and $\psi_{\text{triplet}}(r, y_{o_h}, y_{o_t})$ as follows:

$$\psi_{\text{visual}}(r, I) = \exp\left(-\left(d(\mathbf{T}_r, \mathbf{M}_c z_{ht}) + d(\mathbf{M}_o z_{o_h} + \mathbf{T}_r, \mathbf{M}_o z_{o_t})\right)\right), \tag{6}$$

$$\psi_{\text{triplet}}(r, y_{o_h}, y_{o_t}) = \exp\left(-d(\mathbf{M}_o \mathbf{C}_{y_{o_h}} + \mathbf{T}_r, \mathbf{M}_o \mathbf{C}_{y_{o_t}})\right), \tag{7}$$

where $\mathbf{M}_c \in \mathbb{R}^{K \times D}$ denotes the projection matrix mapping from context space to relation space, and $\mathbf{M}_o \in \mathbb{R}^{K \times D}$ is the projection matrix mapping from object space to relation space. Next, we state how to learn the parameters in JM-SGG model.

## 4.2 Learning

In the learning phase, we seek to learn the parameters $\mathbf{C}$, $\mathbf{T}$, $\mathbf{M}_c$ and $\mathbf{M}_o$ of potential function and the parameters $\theta$ of object detector by maximum likelihood estimation, where $\Theta$ summarizes all these parameters. Specifically, we aim to maximize the expectation of log-likelihood function $\log p_\Theta(G|I)$ with respect to the data distribution $p_d$, *i.e.* $\mathcal{L}(\Theta) = \mathbb{E}_{G \sim p_d}\big[\log p_\Theta(G|I)\big]$, by performing gradient ascent. The gradient of the objective function $\mathcal{L}(\Theta)$ with respect to $\Theta$ can be computed as below:

$$\nabla_\Theta \mathcal{L}(\Theta) = \mathbb{E}_{G \sim p_d}[\nabla_\Theta \log f_\Theta(G, I)] - \mathbb{E}_{G \sim p_\Theta}[\nabla_\Theta \log f_\Theta(G, I)], \tag{8}$$

where $p_\Theta$ is the model distribution that approximates $p_d$ (*i.e.* the conditional distribution $p_\Theta(G|I)$ defined by JM-SGG model). This formula has been broadly adopted in the literature [12, 3, 7], and we provide the proof in supplementary material. In practice, we estimate the first expectation in Eq. (8) with the ground-truth scene graphs in a mini-batch. The estimation of the second expectation in Eq. (8) requires to sample scene graphs from the model distribution, which is nontrivial due to the intractable partition function $Z_\Theta(I)$ that sums over all possible scene graphs. One solution is to run the Markov Chain Monte Carlo (MCMC) sampler, but its computational cost is high, and we therefore use *mean-field variational inference* for more efficient sampling (the detailed scheme is stated in Sec. 4.3).

Instead of fixing the parameters of a pre-trained object detector during learning as in former works [53, 36, 37, 34], we fine-tune the parameters of object detector during maximum likelihood learning. In this way, the detector can extract more precise object and context representations by learning the likelihoods of whole scene graphs. Also, we apply a traditional bounding box regression constraint $\mathcal{L}_{\text{reg}}(\Theta)$ [28] to the detector for preserving its localization capability, and these two learning objectives share the same weight. Next, we introduce the inference scheme for JM-SGG model.

## 4.3 Inference

The inference phase aims to compute the conditional distribution $p_\Theta(G|I)$ defined by JM-SGG model and also sample from it. Exact inference is always infeasible due to the complex structures among the latent variables $y_O$ and $R$ of the scene graph as well as the intractable partition function. Therefore, we approximate $p_\Theta(G|I)$ with a variational distribution $q_\Theta(G)$ via the mean-field approximation [31, 24]:

$$q_\Theta(G) = \prod_{o \in O} q_\Theta(y_o) \prod_{(o_h, r, o_t) \in R} q_\Theta(r), \tag{9}$$

where each factor $q_\Theta(y_o)$ and $q_\Theta(r)$ defines a categorical distribution, *i.e.* $\sum_{y_o \in \mathcal{C}} q_\Theta(y_o) = 1$ and $\sum_{r \in \mathcal{T}} q_\Theta(r) = 1$. In this variational distribution, all object and relation labels are assumed to be independent, and it shares the same set of parameters $\Theta$ with $p_\Theta(G|I)$, which greatly reduces the number of parameters needed for variational inference. For brevity, we will omit $\Theta$ in the following distribution notations, *e.g.* simplifying $q_\Theta(G)$ as $q(G)$.

In general, we are seeking for a variational distribution that satisfies the factorization in Eq. (9) and also maximizes the variational lower bound $\mathcal{L}(q) = \mathbb{E}_{q(G)}[\log p(G, I) - \log q(G)]$ (*i.e.* equivalent to minimizing the KL divergence between $q(G)$ and $p(G|I)$). Typically, this is achieved by optimizing the variational distribution with fixed-point iterations [44, 45], which can however be inefficient, especially for the images with many objects. We thus design an inference algorithm that appropriately initializes each factor in $q(G)$ and iteratively updates all factors. Intuitively, factor initialization is similar to existing SGG methods, where object and relation labels are predicted independently; factor update can be viewed as a refinement procedure, which makes the predictions from the initialization step more consistent. With factor initialization and factor update, the proposed inference method combines the advantages of both existing methods and CRFs, *i.e.* efficiency and consistency.

**Factor initialization.** For initialization, we neglect the interdependency among different object and relation labels, *i.e.* omitting the potential function $\psi_{\text{triplet}}(r, y_{o_h}, y_{o_t})$ in $p(G|I)$, yielding a simplified model distribution $\hat{p}(G|I)$. In this way, we can easily derive the following factors for initialization which makes $q(G) = \hat{p}(G|I)$:

$$q(y_o) = \frac{\phi(y_o, I)}{\sum_{y_o' \in \mathcal{C}} \phi(y_o', I)} \quad \forall o \in O, \tag{10}$$

$$q(r) = \frac{\psi_{\text{visual}}(r, I)}{\sum_{r' \in \mathcal{T}} \psi_{\text{visual}}(r', I)} \quad \forall (o_h, r, o_t) \in R. \tag{11}$$

See supplementary material for the proof. Intuitively, we initialize each factor by only considering its dependency on visual representations, and, on such basis, label interdependency will then be taken into account to refine each factor. In such an initialization approach, the computation of different factors is independent with each other and thus can be done efficiently in a parallel manner. In Sec. 6.1, we empirically illustrate the better convergence performance of this initialization scheme compared to the random initialization which is commonly employed in previous works [45, 22].

**Factor update.** Based on these initialized factors, we perform update by taking into account the interdependency among the object and relation labels in scene graph, *i.e.* using the full expression of $p(G|I)$ with potential function $\psi_{\text{triplet}}(r, y_{o_h}, y_{o_t})$. In the mean-field formulation of Eq. (9), if we are to update one factor $q(y_o)$ (or $q(r)$) with all other factors fixed, its optimum $q^*(y_o)$ (or $q^*(r)$) which maximizes the variational lower bound $\mathcal{L}(q)$ can be specified by the following expression:

$$
\begin{aligned}
\log q^*(y_o) = \log \phi(y_o, I) + \sum_{(o,r,o_t) \in R} \sum_{y_{o_t} \in \mathcal{C}} \sum_{r \in \mathcal{T}} q(y_{o_t}) q(r) \log \psi_{\text{triplet}}(r, y_o, y_{o_t}) \\
+ \sum_{(o_h, r, o) \in R} \sum_{y_{o_h} \in \mathcal{C}} \sum_{r \in \mathcal{T}} q(y_{o_h}) q(r) \log \psi_{\text{triplet}}(r, y_{o_h}, y_o) + \text{const} \quad \forall o \in O,
\end{aligned}
\tag{12}
$$

$$
\begin{aligned}
\log q^*(r) = \log \psi_{\text{visual}}(r, I) \\
+ \sum_{y_{o_h} \in \mathcal{C}} \sum_{y_{o_t} \in \mathcal{C}} q(y_{o_h}) q(y_{o_t}) \log \psi_{\text{triplet}}(r, y_{o_h}, y_{o_t}) + \text{const} \quad \forall (o_h, r, o_t) \in R.
\end{aligned}
\tag{13}
$$

The proof is provided in supplementary material. During computation, we omit the additive constants above, since they can be naturally eliminated when computing normalized $q^*(y_o)$ and $q^*(r)$, *i.e.* taking the exponential of both sides and normalizing $q^*(y_o)$ over $\mathcal{C}$ and $q^*(r)$ over $\mathcal{T}$. Taking a close look at Eqs. (12) and (13), we can find that each factor is updated by aggregating the information from its neighboring factors (*e.g.* from the factors $q(y_{o_h})$ and $q(y_{o_t})$ of head and tail objects to the factor $q(r)$ of their relation), which can be efficiently implemented by matrix multiplication as in message passing neural networks [8]. In practice, we simultaneously update all factors in a single iteration based on the states of factors in last iteration, *i.e.* performing asynchronous message passing in mean field [41, 47], which forms an efficient iterative update scheme. We analyze the efficiency and efficacy of this update scheme in Secs. 6.1 and 6.2.

**Inference algorithm.** The whole inference algorithm is summarized in Alg. 1. Upon on the input scene image $I$, we first initialize each factor in $q(G)$ by Eqs. (10) and (11). After that, we perform factor update for $N_T$ iterations. In each iteration, the log-optimum of each factor is computed based on the factors of last iteration by Eqs. (12) and (13), and the normalized factors are then derived by softmax for update.

---

**Algorithm 1** Inference algorithm of JM-SGG.

**Input:** Scene image $I$, iteration number $N_T$.
**Output:** Factors $\{q(y_o)\}$, $\{q(r)\}$ of $q(G)$.
Initialize $\{q(y_o)\}$, $\{q(r)\}$ by Eqs. (10), (11).
**for** $t = 1$ **to** $N_T$ **do**
  Derive $\{\log q^*(y_o)\}$, $\{\log q^*(r)\}$ by Eqs. (12), (13).
  Update all factors:
    $\{q(y_o)\} \leftarrow \{\text{softmax}(\log q^*(y_o))\}$,
    $\{q(r)\} \leftarrow \{\text{softmax}(\log q^*(r))\}$.
**end for**

---

**Sampling strategy.** After such an iterative inference, we obtain a factorized variational distribution $q(G)$ which well approximates the conditional distribution $p(G|I)$ defined by JM-SGG model. Now, instead of sampling from the intractable model distribution $p(G|I)$, we can easily sample scene graphs from $q(G)$ by independently drawing each object/relation label from the corresponding factor (*i.e.* $q(y_o)$ or $q(r)$), where each factor is a categorical distribution. In practice, we sample $N_S$ scene graphs from $q(G)$ for each image in a mini-batch, yielding totally $N_S N_B$ samples for estimating the second expectation term in $\nabla_\Theta \mathcal{L}(\Theta)$ (Eq. (8)), where $N_B$ denotes batch size.

**Prediction strategy.** At the test time, we need to infer the scene graph with the highest probability in $p(G|I)$, and it can also be efficiently done using the variational distribution $q(G)$. In specific, based on the factorized definition of $q(G)$, we can easily select the object category (or relation type) with the highest probability in each factor $q(y_o)$ (or $q(r)$), and the selected object and relation labels together form a scene graph that well approximates the most likely scene graph with respect to the model distribution $p(G|I)$. Similar prediction strategies have been widely used in previous works that employed mean-field methods [15, 40].

# 5 Experiments

## 5.1 Experimental Setup

**Dataset.** We use the Visual Genome (VG) dataset [16] (CC BY 4.0 License), a large-scale database with structured image concepts, for evaluation. We use the pre-processed VG from Xu et al. [48] (MIT License) which contains 108k images with 150 object categories and 50 relation types. Following previous works [53, 36, 37], we employ the original split with 70% images for training and 30% images for test, and 5k images randomly sampled from the training split are held out for validation.

**Evaluation tasks.** We evaluate the proposed method on two widely studied tasks:

- **Relationship Retrieval (RR).** This task examines model's comprehensive capability of localizing and classifying objects and their relationships. It is further divided into three sub-tasks from easy to hard: (1) Predicate Classification (*PredCls*): predict the predicate/relation of all object pairs using the ground-truth bounding boxes and object labels; (2) Scene Graph Classification (*SGCls*): predict all object categories and relation types given the ground-truth bounding boxes; (3) Scene Graph Generation (SGGen): localize the objects in an image and simultaneously predict their categories and all relations, where an object is regarded as correctly detected if it has at least 0.5 IoU overlap with the ground-truth box. Since two evaluation protocols were typically used in the literature, we adopt two metrics in our experiments, *i.e.* computing the recall for each relation type and reporting the mean ($mR@k$) [21, 48, 53] and computing a single recall for all relation types ($R@k$) [4, 37, 34], where we use both 50 and 100 for $k$ as in previous works [21, 48, 4]. Following Xu et al. [48], we apply the *graph constraint* that only one relation is obtained for each ordered object pair. Totally, we report model's performance on 12 configurations.

- **Zero-Shot Relationship Retrieval (ZSRR).** This task was first introduced by Lu et al. [21] to evaluate model's ability of identifying the head-relation-tail triplets that have not been observed during training. For this task, we employ the metric Zero-Shot Recall@$k$ ($ZSR@k$) and conduct evaluation under three settings, *i.e.* PredCls, SGCls and SGGen. Also, the configurations where $k$ equals to 50 and 100 are both evaluated.

**Performance comparisons.** We compare the proposed method with existing scene graph generation algorithms, including IMP+ [48] (a re-implementation of IMP by Zellers et al. [53]), VTransE [55], FREQ [53], Motifs [53], KERN [4], VCTree [36], VCTree-TDE [37], VCTree-EBM [34] and GB-Net-$\beta$ [52]. We adapt the results on the metric $mR@k$ from original papers, and the results on the metric $R@k$ and $ZSR@k$ are evaluated by the released source code for some methods, *i.e.* VTransE, VCTree-TDE and VCTree-EBM on $R@k$, and KERN and GB-Net-$\beta$ on $ZSR@k$.

## 5.2 Implementation Details

**Model details.** Following previous works [48, 55, 53, 4, 52], we adopt the Faster R-CNN [28] with a VGG-16 [32] backbone as object detector, and the VGG-16 backbone is initialized with the weights of the model pre-trained on ImageNet [30]. We use the same detector configuration as Zellers et al. [53] for fair comparison. The dimension $D$ of object and context space and the dimension $K$ of relation space are both set as 4096, *i.e.* the output dimension of the fc7 layer of VGG-16. Our method is implemented under PyTorch [25], and the source code will be released for reproducibility.

**Training details.** In our experiments, the object detector is first pre-trained by an SGD optimizer (batch size: 4, initial learning rate: 0.001, momentum: 0.9, weight decay: $5 \times 10^{-4}$) for 20 epochs, and the learning rate is multiplied by 0.1 after the 10th epoch. During maximum likelihood learning, we train the potential functions and fine-tune the object detector with another SGD optimizer (batch size: 4, potential function learning rate: 0.001, detector learning rate: 0.0001, momentum: 0.9, weight decay: $5 \times 10^{-4}$) for 10 epochs, and the learning rate is multiplied by 0.1 after the 5th epoch. Without otherwise specified, the iteration number $N_T$ is set as 1 for training and 2 for test, and the per image sampling size $N_S$ is set as 3. These hyperparameters are selected by the grid search on validation set, and their sensitivities are analyzed in Sec. 6.2. An NVIDIA Tesla V100 GPU is used for training.

**Evaluation details.** As stated in Sec. 4.3, we independently predict each object category and relation type by selecting the most likely one in the corresponding factor of variational distribution. The objects predicted as "background" are discarded along with the relations linking to them, and the relations predicted as "no relation" are also removed. To derive a ranked triplet list for RR and ZSRR tasks, we save the probability of each object and relation and compute the probability product

Table 1: Relationship Retrieval performance of various methods in terms of R@$k$ and mR@$k$.

| Task | PredCls | | | | SGCls | | | | SGGen | | | |
|---|---|---|---|---|---|---|---|---|---|---|---|---|
| Method | mR@50 | mR@100 | R@50 | R@100 | mR@50 | mR@100 | R@50 | R@100 | mR@50 | mR@100 | R@50 | R@100 |
| IMP+ [48] | 9.8 | 10.5 | 59.3 | 61.3 | 5.8 | 6.0 | 34.6 | 35.4 | 3.8 | 4.8 | 20.7 | 24.5 |
| VTransE [55] | 17.1 | 18.6 | 60.7 | 62.1 | 8.2 | 8.7 | 35.0 | 35.7 | 6.8 | 8.0 | 22.3 | 25.9 |
| FREQ [53] | 13.3 | 15.8 | 59.9 | 64.1 | 6.8 | 7.8 | 32.4 | 34.0 | 4.3 | 5.6 | 23.5 | 27.6 |
| Motifs [53] | 13.3 | 14.4 | 65.2 | 67.1 | 7.1 | 7.6 | 35.8 | 36.5 | 5.3 | 6.1 | 27.2 | 30.3 |
| KERN [4] | 17.7 | 19.2 | 65.8 | 67.6 | 9.4 | 10.0 | 36.7 | 37.4 | 6.4 | 7.3 | 27.1 | 29.8 |
| VCTree [36] | 17.9 | 19.4 | 66.4 | 68.1 | 10.1 | 10.8 | 38.1 | 38.8 | 6.9 | 8.0 | 27.9 | 31.3 |
| VCTree-TDE [37] | **25.4** | **28.7** | 67.0 | 68.7 | 12.2 | 14.0 | 39.2 | 40.3 | 9.3 | 11.1 | 28.2 | 31.5 |
| VCTree-EBM [34] | 18.2 | 19.7 | 66.8 | 68.3 | 12.5 | 13.5 | 38.7 | 39.2 | 7.7 | 9.1 | 28.0 | 31.3 |
| GB-Net-$\beta$ [52] | 22.1 | 24.0 | 66.6 | 68.2 | 12.7 | 13.4 | 37.3 | 38.0 | 7.1 | 8.5 | 26.3 | 29.9 |
| JM-SGG (triplet) | 23.0 | 25.6 | 68.5 | 69.3 | 11.6 | 13.2 | 41.9 | 42.5 | 8.4 | 10.6 | 29.0 | 31.2 |
| JM-SGG (w/o FU) | 23.7 | 26.5 | 69.1 | 70.2 | 12.1 | 13.6 | 42.7 | 43.2 | 8.7 | 10.5 | 28.2 | 31.3 |
| JM-SGG | 24.9 | 28.0 | **70.8** | **71.7** | **13.1** | **14.7** | **43.4** | **44.2** | **9.8** | **11.8** | **29.3** | **32.2** |

within each head-relation-tail triplet, and all triplets are then ranked according to the values of their probability products in a descending order. We report model's performance at the last epoch.

## 5.3  Experimental Results

**Relationship Retrieval (RR).** In Tab. 1, we compare our method with existing approaches under 12 settings of the RR task. It can be observed that the proposed JM-SGG model achieves the best performance on 10 of 12 settings. In particular, compared to the state-of-the-art VCTree-TDE [37], a previous work dedicated to addressing unbiased scene graph prediction, JM-SGG performs better on 4 of 6 settings for unbiased prediction (*i.e.* the settings using metric mR@$k$). We think these superior results are mainly ascribed to the proposed joint scene graph modeling, in which the class imbalance among different relation types is mitigated by emphasizing the role of these sample-scarce relation types under the context of whole scene graphs.

**Zero-Shot Relationship Retrieval (ZSRR).** Tab. 2 reports the performance of various approaches on 6 settings of the ZSRR task. The comparison with FREQ [53] is not included on this task, since this baseline method can only predict the relational triplets appearing in the training set. We can observe that the JM-SGG model outperforms existing methods on all 6 settings, and, especially, a 34% performance gain on ZSR@50 is achieved on the SGCls sub-task. These results illustrate the effectiveness of JM-SGG on discovering the novel relational triplets that have not been observed during learning.

Table 2: Zero-Shot Relationship Retrieval performance of various methods in terms of ZSR@$k$.

| Task | PredCls | | SGCls | | SGGen | |
|---|---|---|---|---|---|---|
| Method | @50 | @100 | @50 | @100 | @50 | @100 |
| IMP+ [48] | 12.6 | 15.2 | 3.3 | 3.8 | 0.5 | 1.2 |
| VTransE [55] | 11.3 | 14.7 | 2.5 | 3.3 | 0.8 | 1.5 |
| Motifs [53] | 10.9 | 14.5 | 2.2 | 3.0 | 0.1 | 0.2 |
| KERN [4] | 12.3 | 15.4 | 3.2 | 3.8 | 0.6 | 1.4 |
| VCTree [36] | 10.8 | 14.3 | 1.9 | 2.6 | 0.2 | 0.7 |
| VCTree-TDE [37] | 14.3 | 17.6 | 3.2 | 4.0 | 2.6 | 3.2 |
| VCTree-EBM [34] | 11.6 | 15.7 | 3.8 | 5.1 | 2.5 | 3.4 |
| GB-Net-$\beta$ [52] | 12.8 | 16.0 | 3.4 | 4.1 | 2.0 | 2.6 |
| JM-SGG (triplet) | 12.7 | 16.3 | 3.7 | 4.8 | 2.0 | 3.1 |
| JM-SGG (w/o FU) | 14.2 | 18.0 | 4.2 | 5.5 | 2.3 | 3.2 |
| JM-SGG | **14.9** | **18.6** | **5.1** | **6.2** | **2.9** | **3.6** |

## 6  Analysis

### 6.1  Ablation Study

**Ablation study for joint scene graph modeling.** To better verify the effectiveness of joint scene graph modeling, we study a variant of JM-SGG which models the joint distribution of an individual relational triplet instead of the whole scene graph, denoted as *JM-SGG (triplet)* (see supplementary material for more details). In Tabs. 1 and 2, JM-SGG clearly outperforms JM-SGG (triplet) on all metrics including the metric mR@$k$ for unbiased prediction, which demonstrates the benefit of joint scene graph modeling on mitigating the class imbalance among different relation types.

**Ablation study for factor initialization.** In this experiment, we compare the proposed initialization method (Eqs. (10) and (11)) with the random initialization which randomly initializes the categorical distribution for each factor $q(y_o)$ and $q(r)$ in variational distribution $q(G)$. Under these two initialization schemes, we respectively plot model's performance after different iterations of factor update in Fig. 2(a). After four iterations, two schemes converge to the solutions with comparable performance, while our initialization approach shows a faster convergence (*i.e.* converge after two iterations).

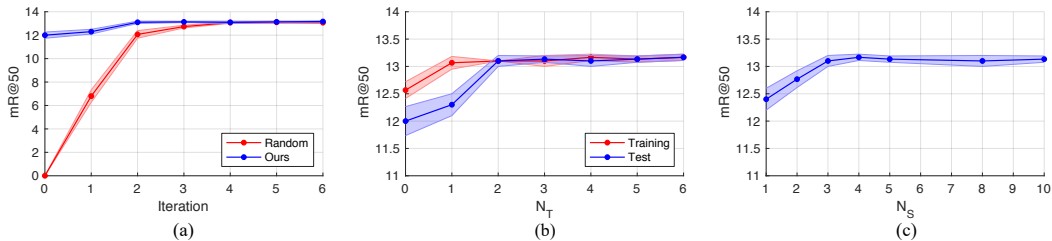

Figure 2: (a) Comparison between random initialization and the proposed initialization scheme. (b) Sensitivity analysis of iteration number $N_T$ during training and test. (c) Sensitivity analysis of per image sampling size $N_S$. (All results are reported on SGCls task under the metric mR@50.)

Table 3: Ablation study on modeling head-relation-tail triplets.

| Task | PredCls | | | | SGCls | | | | SGGen | | | |
|---|---|---|---|---|---|---|---|---|---|---|---|---|
| Method | mR@50 | mR@100 | R@50 | R@100 | mR@50 | mR@100 | R@50 | R@100 | mR@50 | mR@100 | R@50 | R@100 |
| JM-SGG (TransE) | 22.7 | 25.4 | 68.1 | 68.8 | 11.5 | 13.2 | 41.7 | 42.2 | 7.9 | 9.6 | 27.6 | 29.8 |
| JM-SGG | **24.9** | **28.0** | **70.8** | **71.7** | **13.1** | **14.7** | **43.4** | **44.2** | **9.8** | **11.8** | **29.3** | **32.2** |

**Ablation study for factor update.** In this part, we study another configuration where the initialized factors are directly used for scene graph prediction without factor update, denoted as *JM-SGG (w/o FU)*. In Tabs. 1 and 2, the superior performance of JM-SGG over JM-SGG (w/o FU) verifies the necessity of performing factor update to refine the initial label predictions.

**Ablation study on modeling head-relation-tail triplets.** Previous works [55, 5] used TransE [2] to model the relation between two objects, while our method employs TransR [20] to model head-relation-tail triplets. To investigate the effectiveness of such a model design, we substitute TransR with TransE in our model, named as *JM-SGG (TransE)*. Specifically, this model variant regards object and relation embeddings lie in the same space, and thus the projection matrix $\mathbf{M}_o$ is removed from two relation potential terms $\psi_{\text{visual}}$ and $\psi_{\text{triplet}}$. In Tab. 3, it can be observed that TransR clearly outperforms TransE in the JM-SGG model, which demonstrates the importance of modeling objects and relations in two distinct embedding spaces.

## 6.2 Sensitivity Analysis

**Sensitivity of iteration number $N_T$.** In Fig. 2(b), we plot the performance of JM-SGG model under different iteration numbers. It can be observed that, for training, one iteration of factor update is enough to derive a decent variational distribution for the sampling purpose; for test, two iterations are required to converge to the optimal approximation of the model distribution.

**Sensitivity of per image sampling size $N_S$.** We vary the value of per image sampling size $N_S$ for learning and plot the corresponding model performance in Fig. 2(c). We can observe that through sampling at least three scene graphs from the variational distribution for each image, the second expectation term in Eq. (8) can be well estimated, which stably enhances model performance.

## 6.3 Visualization

In Fig. 3, we visualize the typical scene graphs generated by JM-SGG model, in which the results with and without applying factor update are respectively shown. In these two examples, factor update succeeds in correcting some wrong relation labels (*e.g.* person *has* jean $\rightarrow$ person *wearing* jean) by considering the dependency among different object and relation labels. More visualization results are provided in the supplementary material.

## 7 Conclusions and Future Work

In this work, we propose the Joint Modeling for Scene Graph Generation (JM-SGG) model. This model is able to jointly capture the dependency among all object and relation labels in the scene graph, and its learning and inference can be efficiently performed using the mean-field variational inference

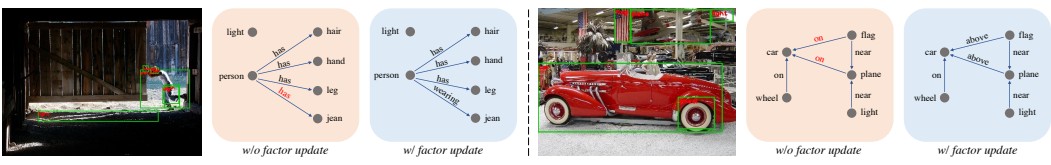

Figure 3: The scene graphs generated by JM-SGG model. (*Red* labels in scene graphs are wrong.)

algorithm. The extensive experiments on both relationship retrieval and zero-shot relationship retrieval tasks demonstrate the superiority of JM-SGG model.

The current JM-SGG model cannot be directly used for visual reasoning, and its inference method makes a strong assumption of fully factorized variational distribution. Therefore, our future work will include exploring downstream visual reasoning tasks (*e.g.* visual question answering [1] and visual commonsense reasoning [54]) based on JM-SGG model and further improving our approximate inference algorithm (*e.g.* by defining more expressive variational distribution).

## 8   Broader Impacts

This research project focuses on predicting objects and their relations in a visual scene by fully capturing the dependency among all objects and relations, and the predicted object and relation labels are further organized as a scene graph. Compared to the conventional visual recognition systems that only predict objects, our approach is able to simultaneously provide object and relationship prediction. This merit enables more in-depth scene understanding and can potentially benefit many real-world applications, like intelligent surveillance and autonomous driving.

However, it cannot be denied that the annotation process for a scene graph generation model is labor-intensive. For example, 11.5 objects and 6.2 relations, on average, are required to be annotated for each image in the Visual Genome dataset, and the dataset contains 108k images in total. Therefore, how to train a scene graph generation model in a more efficient way by using less labeled data remains to be further explored.

## Acknowledgments and Disclosure of Funding

This project was supported by the Natural Sciences and Engineering Research Council (NSERC) Discovery Grant, the Canada CIFAR AI Chair Program, collaboration grants between Microsoft Research and Mila, Samsung Electronics Co., Ldt., Amazon Faculty Research Award, Tencent AI Lab Rhino-Bird Gift Fund and a NRC Collaborative R&D Project (AI4D-CORE-08). This project was also partially funded by IVADO Fundamental Research Project grant PRF2019-3583139727. Bingbing Ni is supported by National Science Foundation of China (U20B2072, 61976137).

The authors would like to thank Zhaocheng Zhu, Louis-Pascal Xhonneux and Zuobai Zhang for providing constructive advices during this project, and also appreciate the Student Innovation Center of SJTU for providing GPUs.

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
