# Joint Modeling of Visual Objects and Relations for Scene Graph Generation (*Supplementary Material*)

**Minghao Xu**[1]  **Meng Qu**[2,3]  **Bingbing Ni**[1]  **Jian Tang**[2,4,5]
[1]Shanghai Jiao Tong University, Shanghai 200240, China  [2]Mila - Québec AI Institute
[3]University of Montréal  [4]HEC Montréal  [5]CIFAR AI Research Chair
{xuminghao118, nibingbing}@sjtu.edu.cn  meng.qu@umontreal.ca  jian.tang@hec.ca

## 1 Proofs

**Theorem 1.** *The gradient of log-likelihood function $\mathcal{L}(\Theta) = \mathbb{E}_{G \sim p_d}\big[\log p_\Theta(G|I)\big]$ can be computed with the unnormalized likelihood function $f_\Theta$ as below:*
$$\nabla_\Theta \mathcal{L}(\Theta) = \mathbb{E}_{G \sim p_d}[\nabla_\Theta \log f_\Theta(G, I)] - \mathbb{E}_{G \sim p_\Theta}[\nabla_\Theta \log f_\Theta(G, I)],$$
*where $p_d$ is the data distribution, $p_\Theta$ denotes the conditional distribution $p_\Theta(G|I)$ defined by the model.*

*Proof.* Based on the formulation of the likelihood function $p_\Theta(G|I) = f_\Theta(G, I)/Z_\Theta(I)$, we can reformulate the gradient of log-likelihood function as:
$$\nabla_\Theta \mathcal{L}(\Theta) = \mathbb{E}_{G \sim p_d}[\nabla_\Theta \log f_\Theta(G, I)] - \nabla_\Theta \log Z_\Theta(I).$$
For the second term of this equation, we deduce as below:
$$\begin{aligned}
\nabla_\Theta \log Z_\Theta(I) &= \frac{1}{Z_\Theta(I)} \nabla_\Theta Z_\Theta(I) \\
&= \frac{1}{Z_\Theta(I)} \nabla_\Theta \sum_{G \in \mathbb{G}} f_\Theta(G, I) \\
&= \frac{1}{Z_\Theta(I)} \sum_{G \in \mathbb{G}} \nabla_\Theta f_\Theta(G, I) \\
&= \frac{1}{Z_\Theta(I)} \sum_{G \in \mathbb{G}} f_\Theta(G, I) \nabla_\Theta \log f_\Theta(G, I) \\
&= \sum_{G \in \mathbb{G}} p_\Theta(G|I) \nabla_\Theta \log f_\Theta(G, I) \\
&= \mathbb{E}_{G \sim p_\Theta}[\nabla_\Theta \log f_\Theta(G, I)],
\end{aligned}$$
where $\mathbb{G}$ is the set of all possible scene graphs for image $I$. In conclusion, we can express $\nabla_\Theta \mathcal{L}(\Theta)$ as below:
$$\nabla_\Theta \mathcal{L}(\Theta) = \mathbb{E}_{G \sim p_d}[\nabla_\Theta \log f_\Theta(G, I)] - \mathbb{E}_{G \sim p_\Theta}[\nabla_\Theta \log f_\Theta(G, I)].$$
$\square$

**Theorem 2.** *In the initialization phase, the potential function $\psi_{\text{triplet}}(r, y_{o_h}, y_{o_t})$ for modeling label dependency is omitted in $p(G|I)$, yielding a simplified model distribution $\hat{p}(G|I)$. The following factors make the variational distribution $q(G)$ equal to $\hat{p}(G|I)$:*
$$q(y_o) = \frac{\phi(y_o, I)}{\sum_{y'_o \in \mathcal{C}} \phi(y'_o, I)} \quad \forall o \in O, \quad q(r) = \frac{\psi_{\text{visual}}(r, I)}{\sum_{r' \in \mathcal{T}} \psi_{\text{visual}}(r', I)} \quad \forall (o_h, r, o_t) \in R.$$

35th Conference on Neural Information Processing Systems (NeurIPS 2021).

*Proof.* By not considering the potential function $\psi_{\text{triplet}}(r, y_{o_h}, y_{o_t})$, the simplified model distribution $\hat{p}(G|I)$ has the following form:

$$\hat{p}(G|I) = \frac{1}{Z_\Theta(I)} \prod_{o \in O} \phi(y_o, I) \prod_{(o_h, r, o_t) \in R} \psi_{\text{visual}}(r, I).$$

If the factors $q(y_o)$ ($o \in O$) and $q(r)$ ($(o_h, r, o_t) \in R$) take the expressions in Theorem 1, we can express the variational distribution $q(G)$ as below:

$$q(G) = \prod_{o \in O} \frac{\phi(y_o, I)}{\sum_{y'_o \in \mathcal{C}} \phi(y'_o, I)} \prod_{(o_h, r, o_t) \in R} \frac{\psi_{\text{visual}}(r, I)}{\sum_{r' \in \mathcal{T}} \psi_{\text{visual}}(r', I)}.$$

In this situation, $q(G)$ and $\hat{p}(G|I)$ take a very similar expression. We further express the partition function $Z_\Theta(I)$ with the following form:

$$Z_\Theta(I) = \prod_{o \in O} \sum_{y'_o \in \mathcal{C}} \phi(y'_o, I) \prod_{(o_h, r, o_t) \in R} \sum_{r' \in \mathcal{T}} \psi_{\text{visual}}(r', I).$$

Now, we can exactly derive that $q(G) = \hat{p}(G|I)$.

$\square$

**Theorem 3.** *In the update phase, we use the full expression of $p(G|I)$ with the potential function $\psi_{\text{triplet}}(r, y_{o_h}, y_{o_t})$ for modeling label dependency. In the variational distribution $q(G)$, if we are to update one factor $q(y_o)$ (or $q(r)$) with all other factors fixed, its optimum $q^*(y_o)$ (or $q^*(r)$) which maximizes the variational lower bound $\mathcal{L}(q) = \mathbb{E}_{q(G)}[\log p(G, I) - \log q(G)]$ can be specified by the following expression:*

$$\log q^*(y_o) = \log \phi(y_o, I) + \sum_{(o, r, o_t) \in R} \sum_{y_{o_t} \in \mathcal{C}} \sum_{r \in \mathcal{T}} q(y_{o_t}) q(r) \log \psi_{\text{triplet}}(r, y_o, y_{o_t})$$
$$+ \sum_{(o_h, r, o) \in R} \sum_{y_{o_h} \in \mathcal{C}} \sum_{r \in \mathcal{T}} q(y_{o_h}) q(r) \log \psi_{\text{triplet}}(r, y_{o_h}, y_o) + \text{const} \quad \forall o \in O,$$

$$\log q^*(r) = \log \psi_{\text{visual}}(r, I)$$
$$+ \sum_{y_{o_h} \in \mathcal{C}} \sum_{y_{o_t} \in \mathcal{C}} q(y_{o_h}) q(y_{o_t}) \log \psi_{\text{triplet}}(r, y_{o_h}, y_{o_t}) + \text{const} \quad \forall (o_h, r, o_t) \in R.$$

*Proof.* By substituting the formulation of the assumed variational distribution into the variational lower bound $\mathcal{L}(q)$, we can have the following expression:

$$\mathcal{L}(q) = \sum_G \left\{ \prod_{o \in O} q(y_o) \prod_{(o_h, r, o_t) \in R} q(r) \left[ \log p(G, I) - \sum_{o \in O} \log q(y_o) - \sum_{(o_h, r, o_t) \in R} \log q(r) \right] \right\}.$$

(1) For updating the factor $q(y_o)$ ($o \in O$) with all other factors fixed, we can deduce the variational lower bound as below:

$$\mathcal{L}(q) = \sum_{y_o \in \mathcal{C}} q(y_o) \sum_{G \setminus \{o\}} \left\{ \prod_{o' \in O \setminus \{o\}} q(y_{o'}) \prod_{(o_h, r, o_t) \in R} q(r) \log p(G, I) \right\}$$
$$- \sum_{y_o \in \mathcal{C}} q(y_o) \log q(y_o) + \text{const}$$
$$= -D_{\text{KL}} \big[ q(y_o) \,||\, \tilde{p}(y_o, I) \big] + \text{const},$$

$$\log \tilde{p}(y_o, I) = \sum_{G \setminus \{o\}} \left\{ \prod_{o' \in O \setminus \{o\}} q(y_{o'}) \prod_{(o_h, r, o_t) \in R} q(r) \log p(G, I) \right\} + \text{const},$$

where $D_{\mathrm{KL}}$ denotes the KL divergence. In this case, maximizing $\mathcal{L}(q)$ is equivalent to minimizing the KL divergence term, and the minimum occurs when $q(y_o) = \tilde{p}(y_o, I)$. We thus get the expression of the optimal solution $q^*(y_o)$ as below:

$$
\begin{aligned}
\log q^*(y_o) &= \log \tilde{p}(y_o, I) \\
&= \sum_{G \backslash \{o\}} \left\{ \prod_{o' \in O \backslash \{o\}} q(y_{o'}) \prod_{(o_h, r, o_t) \in R} q(r) \log p(G, I) \right\} + \mathrm{const} \\
&= \log \phi(y_o, I) + \sum_{(o, r, o_t) \in R} \sum_{y_{o_t} \in \mathcal{C}} \sum_{r \in \mathcal{T}} q(y_{o_t}) q(r) \log \psi_{\mathrm{triplet}}(r, y_o, y_{o_t}) \\
&\quad + \sum_{(o_h, r, o) \in R} \sum_{y_{o_h} \in \mathcal{C}} \sum_{r \in \mathcal{T}} q(y_{o_h}) q(r) \log \psi_{\mathrm{triplet}}(r, y_{o_h}, y_o) + \mathrm{const}.
\end{aligned}
$$

(2) For updating the factor $q(r)$ $((o_h, r, o_t) \in R)$ with all other factors fixed, we can deduce the variational lower bound as below:

$$
\begin{aligned}
\mathcal{L}(q) &= \sum_{r \in \mathcal{T}} q(r) \sum_{G \backslash \{(o_h, r, o_t)\}} \left\{ \prod_{o \in O} q(y_o) \prod_{(o_h', r', o_t') \in R \backslash \{(o_h, r, o_t)\}} q(r') \log p(G, I) \right\} \\
&\quad - \sum_{r \in \mathcal{T}} q(r) \log q(r) + \mathrm{const} \\
&= -D_{\mathrm{KL}}\big[ q(r) \,\|\, \tilde{p}(r, I) \big] + \mathrm{const},
\end{aligned}
$$

$$
\log \tilde{p}(r, I) = \sum_{G \backslash \{(o_h, r, o_t)\}} \left\{ \prod_{o \in O} q(y_o) \prod_{(o_h', r', o_t') \in R \backslash \{(o_h, r, o_t)\}} q(r') \log p(G, I) \right\} + \mathrm{const}.
$$

Similarly, the maximum of $\mathcal{L}(q)$ is achieved when $q(r) = \tilde{p}(r, I)$, which derives the optimal solution $q^*(r)$ as below:

$$
\begin{aligned}
\log q^*(r) &= \log \tilde{p}(r, I) \\
&= \sum_{G \backslash \{(o_h, r, o_t)\}} \left\{ \prod_{o \in O} q(y_o) \prod_{(o_h', r', o_t') \in R \backslash \{(o_h, r, o_t)\}} q(r') \log p(G, I) \right\} + \mathrm{const} \\
&= \log \psi_{\mathrm{visual}}(r, I) + \sum_{y_{o_h} \in \mathcal{C}} \sum_{y_{o_t} \in \mathcal{C}} q(y_{o_h}) q(y_{o_t}) \log \psi_{\mathrm{triplet}}(r, y_{o_h}, y_{o_t}) + \mathrm{const}.
\end{aligned}
$$

$\square$

## 2 More Implementation Details

**Implementation details of JM-SGG (triplet).** In the ablation study, we study a variant of JM-SGG that models the joint distribution of an individual relational triplet instead of the whole scene graph, named as JM-SGG (triplet), which has the same model expressiveness as two previous works [2, 1]. For a relational triplet $e$ composed of head object $o_h$, tail object $o_t$ and relation $r$, JM-SGG (triplet) models the joint distribution of all possible triplet labels upon image $I$ as below:

$$
p_\Theta(e|I) = \frac{1}{Z_\Theta(I)} f_\Theta(e, I), \tag{1}
$$

$$
f_\Theta(e, I) = \phi(y_{o_h}, I) \, \phi(y_{o_t}, I) \, \psi(r, y_{o_h}, y_{o_t}, I), \tag{2}
$$

where $\Theta$ summarizes all model parameters, $f_\Theta$ is an unnormalized likelihood function, and $Z_\Theta$ is the partition function. The definitions of potential function $\phi$ and $\psi$ follow those in JM-SGG model.

In the learning phase, we use a similar expression as the one in Theorem 1 to estimate the gradients for maximum likelihood learning:

$$
\nabla_\Theta \mathcal{L}(\Theta) = \mathbb{E}_{e \sim p_d}[\nabla_\Theta \log f_\Theta(e, I)] - \mathbb{E}_{e \sim p_\Theta}[\nabla_\Theta \log f_\Theta(e, I)]. \tag{3}
$$

We employ the triplets in ground-truth scene graphs to estimate the first expectation in Eq. (3), and the triplets sampled from the model distribution are used to estimate the second expectation in Eq. (3).

In the inference phase, JM-SGG (triplet) also defines a variational distribution $q_\Theta(e)$ based on mean-field approximation:

$$q_\Theta(e) = q_\Theta(y_{o_h})q_\Theta(y_{o_t})q_\Theta(r). \tag{4}$$

Following JM-SGG, the three factors in Eq. (4) are first independently initialized and then iteratively updated (the iteration number $N_T$ is set as 1 for both sampling and test).

## 3 More Visualization Results

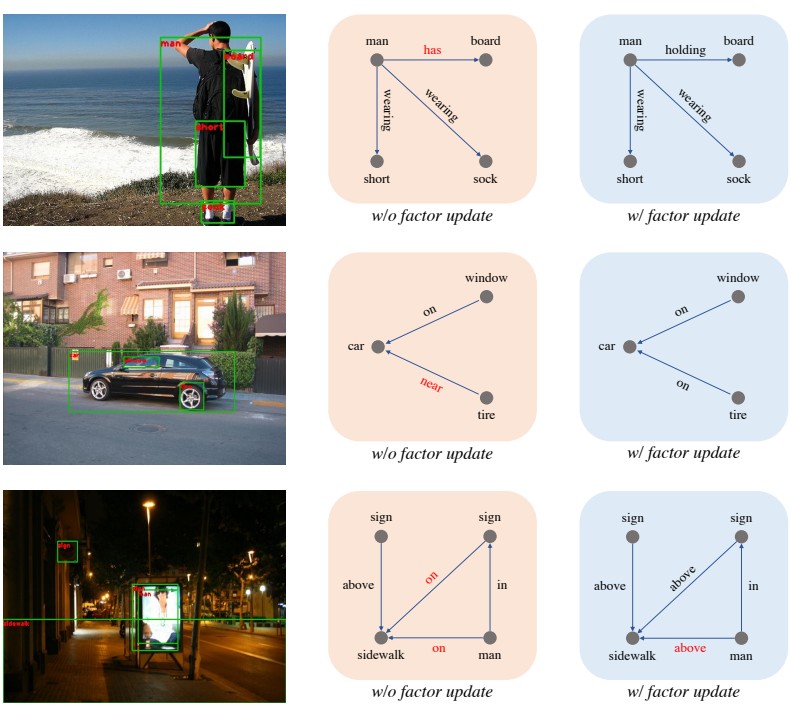

Figure 1: The scene graphs generated by JM-SGG model. (*Red* labels in scene graphs are wrong.)

In Fig. 1, we visualize three typical scene graphs generated by JM-SGG model, in which the results with and without applying factor update are respectively shown. In these examples, factor update is able to correct some wrong relation labels (*e.g.* man *has* board → man *holding* board, sign *on* sidewalk → sign *above* sidewalk) by considering the dependency among different object and relation labels in the scene graph.