# OpenReview forum: "Joint Modeling of Visual Objects and Relations for Scene Graph Generation"
_NeurIPS.cc/2021/Conference — NeurIPS 2021 Poster_

### Official Review · Reviewer_PwTs · 2021-07-15

**Rating:** 7
**Confidence:** 4

**Summary:**

* The paper proposes a method for generating scene graphs of images by modeling the objects and relationships with a Conditional Random Field. The paper claims that other existing methods do not model the dependency between all the objects and relations in an image. The proposed method is meant for modeling all the relations jointly which could lead to better performance in generating scene graphs.
* The potential function for object components is defined using the representation obtained from an object detector network. The potential function for relationship components is defined using the knowledge graph embedding method which represents both the objects and relationships in the same embedding space.
* A mean-field approximation of the probability distribution of the CRF is obtained by treating the labels for objects and relations as independent of each other. A message passing algorithm is used to iteratively update the factors and perform inference with the mean-field approximation. This is also helpful for sampling the scene graphs during the learning phase.
* The experiments are performed on the VisualGenome dataset and results are shown for a few different tasks like Relationship Predicate Classification, Scene Graph Classification and Generation. The paper also includes results on the task of zero shot relationship retrieval that evaluates the model’s ability to identify head-relation-tail combinations not observed during training.


**Limitations And Societal Impact:**

Yes, the authors have provided a short discussion on the limitations and any societal impact the work might have.

**Main Review:**

I have a few concerns regarding the Originality, Quality and Clarity. The concerns are marked with the [-ve] prefix.

Originality
* [-ve] The claim that other works don’t model all the relationships in an image is not true. The idea of using factors for relationships and aggregating information over all the relationships using an iterative message passing algorithm has been used by W.Cong et al. [4]. The Algorithm 1 in [4] has a message passing step that gathers information from the neighbors of a node in a graph, thus aggregating information from the entire graph over the iterations. The aggregation method is not just special in [4], it is also a pretty standard way to do inference over graphs.
* [-ve] The paper claims to propose a novel inference algorithm. However, it is not clear what aspect of the inference algorithm is novel when compared to that of existing mean-field approximation for CRF models. This inference algorithm has been used for CRFs in many prior computer vision works. Ex: Krahenbuhl and Kolton, NeurIPS 2011.

Quality
* [-ve] The results are better than some of the state of the art methods. While the paper claims that this is due to joint scene graph modeling (line 337), I think a more thorough analysis would be required to justify the claim. The effect of the proposed relationship potential function that is different from other works is not fully measured.
* The definitions of the potential functions seem to be the main distinguishing contribution when compared to works like Cong et al. [4]. The work in [4] uses TransE (Bordes et al. 2013) whereas this paper uses TransR [19]. Could this be the one of the reasons for the improved results? This could be verified with an ablation experiment where TransR is substituted with TransE. I am suggesting this experiment because the results from the ablation settings which don’t perform full scene graph modeling (triplet-only and without factor update) are already better in some settings when compared to the other methods.
* The results from the model without factor update is better than the triplet only model which does one step of factor update. This is an interesting result which needs some explanation. Does this imply that the results can be noisy if we don’t perform full scene graph modeling? It will be good to mention this explicitly which adds to the motivation of doing full scene graph modeling.
* The experiments section provides a good sensitivity analysis showing how the results converge with increasing number of factor update iterations.
* The code is provided for reproducing the results

Clarity
* [-ve] The paper is written well to provide a clear explanation of the method along with the proofs of the mathematical lemmas. However, the related work section is lacking since it doesn’t mention how the proposed method is different and/or better compared to existing works.

Significance
* The results are better than the state of the art methods in many settings.
* The work is useful for the community as an example for an alternative method to perform scene generation. While the state of the art methods like Dhingra et al., CVPR 2021 show much better performance on the same tasks using Transformers, the paper proposed a viable method that uses CRFs. It is beneficial to the community since it answers the question “How would CRF models perform on the task of scene generation?”

Other suggestions
* The experiments use VGG16 as the backbone for comparison with other methods. However, it will be good to include additional results with Resnet as the backbone since the latest methods have shown improvements when switching from VGG16 to Resnet. Ex: Dhingra et al., CVPR 2021.


**Time Spent Reviewing:**

24

---

> ### Author Response · Authors · 2021-08-10
> **Author Response to Reviewer PwTs**
>
> Thanks for your valuable comments and constructive suggestions! We respond to your questions and concerns as below:
>
> **-Q1:** The claim that other works don't model all the relationships in an image is not true.
>
> **-A1:** Thanks for pointing out this improper claim. Indeed, there are some existing works that have explored joint scene graph modeling, *e.g.* Scene Graph Generation via Conditional Random Fields (SG-CRF) by Cong et al. 2018 and Energy-Based Learning for Scene Graph Generation (EBM-SGG) by Suhail et al. 2021. Compared to these previous efforts, our work seeks to **design a more expressive model that can better capture the dependencies within the entire scene graph**. Specifically, compared to SG-CRF that only models unary and pairwise (*i.e.* head and tail, head and relation, relation and tail) label dependencies, the proposed JM-SGG model additionally models the ternary label dependencies within all head-relation-tail triplets. These ternary terms are inspired by an existing knowledge graph embedding method, TransR, and they introduce strong inductive bias of relationship modeling to the SGG model. Compared to EBM-SGG that only models the dependencies among various object and relation labels, JM-SGG additionally utilizes the unary potential function to model the dependency between each semantic label and its corresponding visual representation. Therefore, JM-SGG can infer each object and relation label based not only on label dependency but also on exact visual grounding. In the revised version, we will **emphasize these previous efforts** and **tone down our contribution** as "a more expressive model for joint scene graph modeling".
>
> **-Q2:** Compared to existing mean-field approximation for CRF models, what aspect of the proposed inference algorithm is novel?
>
> **-A2:** Indeed, as you suggested, the proposed inference algorithm follows the standard mean-field inference method for CRF which has been used in many computer vision works, *e.g.* Efficient Inference in Fully Connected CRFs with Gaussian Edge Potentials by Krahenbuhl and Kolton, 2011. However, all these works only consider unary and binary terms in their energy functions, where developing mean-field algorithms is easier. In contrast, our work also considers a ternary term to better model the dependency of each head-relation-tail triplet, and existing mean-field methods cannot be directly applied here. Therefore, a different mean-field inference algorithm is developed for our approach. Specifically, in our inference algorithm, *factor initialization* is performed only by unary potential terms, and *factor update* is performed by both unary and ternary potential terms. Therefore, the contribution of our inference algorithm is **a principled mean-field inference approach involving the modeling of relational triplets**. We will revise the corresponding claims in the draft to **better reflect the exact contribution of our inference method**.
>
> **-Q3:** The claim that the performance improvement is mainly ascribed to the joint scene graph modeling should be further justified.
>
> **-A3:** Thanks for pointing out this unverified claim. We consider three model configurations to verify how joint scene graph modeling benefits the model. (1) *Independent Modeling for Scene Graph Generation (IM-SGG)*: this model only uses the unary potential function of each object and relation, and it does not perform iterative label updates due to the lack of higher-order potential functions. (2) *JM-SGG (w/o FU)*: this model employs all unary and ternary potential functions during learning, while it does not perform factor update during inference. (3) *JM-SGG*: this model uses all potential functions during both learning and inference, and it performs factor update for inference. From the results in Tabs A, B and C, we can observe that JM-SGG (w/o FU) outperforms IM-SGG with a clear margin, and JM-SGG achieves a less substantial performance gain upon JM-SGG (w/o FU). These results illustrate that the joint scene graph modeling during learning is a critical factor leading to performance improvement, while the inference-time joint modeling via factor update is not that critical.
>
> The results below will be added to the revised draft together with the analysis above.
>
>
> Table A: Predicate Classification performance with graph constraint on Visual Genome (ablation study for joint scene graph modeling).
>
> |Method|mR@50|mR@100|R@50|R@100|
> |:----:|:----:|:----:|:----:|:----:|
> |IM-SGG|13.4|15.2|63.2|65.5|
> |JM-SGG (w/o FU)|23.7|26.5|69.1|70.2|
> |JM-SGG|**24.9**|**28.0**|**70.8**|**71.7**|
>
>
> Table B: Scene Graph Classification performance with graph constraint on Visual Genome (ablation study for joint scene graph modeling).
>
> |Method|mR@50|mR@100|R@50|R@100|
> |:----:|:----:|:----:|:----:|:----:|
> |IM-SGG|7.4|7.8|36.3|37.0|
> |JM-SGG (w/o FU)|12.1|13.6|42.7|43.2|
> |JM-SGG|**13.1**|**14.7**|**43.4**|**44.2**|
>
>
> Table C: Scene Graph Generation performance with graph constraint on Visual Genome (ablation study for joint scene graph modeling).
>
> |Method|mR@50|mR@100|R@50|R@100|
> |:----:|:----:|:----:|:----:|:----:|
> |IM-SGG|5.2|5.9|24.8|27.1|
> |JM-SGG (w/o FU)|8.7|10.5|28.2|31.3|
> |JM-SGG|**9.8**|**11.8**|**29.3**|**32.2**|
>
>
>
> **-Q4:** The effect of the proposed relationship potential function that is different from other works is not fully explored. In particular, could the use of TransR instead of TransE be one of the reasons for improved performance?
>
> **-A4:** Thanks for this great suggestion. The proposed relation potential function $\psi$ (Eq. (5)) consists of two terms, a visual influence term $\psi_{visual}$ (Eq. (6)) and a triplet-level label consistency term $\psi_{triplet}$ (Eq. (7)), in which we utilize TransR to model the ternary correlation among head object, tail object and their relationship. By comparison, the previous work, Scene Graph Generation via Conditional Random Fields (SG-CRF) by Cong et al. 2018, employed TransE to define the visual influence term (Eq. (4) in their paper), and they modeled pairwise label consistency for message passing (Algorithm 1 in their paper). Therefore, we use two ablation studies to evaluate two design choices of the proposed relation potential function: (1) TransR *vs.* TransE and (2) Ternary label consistency *vs.* Pairwise label consistency.
>
> **TransR *vs.* TransE.** We substitute TransR with TransE to define the relation potential function and name this model variant as *JM-SGG (TransE)*. Specifically, this model regards object and relation embeddings lie in the same space, and thus the projection matrix $M_o$ is removed from two relation potential terms $\psi_{visual}$ and $\psi_{triplet}$. This model is compared with the standard JM-SGG model which is based on TransR. In Tabs. D, E and F, it can be observed that TransR clearly outperforms TransE in the JM-SGG model, which demonstrates the importance of modeling objects and relations in two distinct embedding spaces.
>
> **Ternary label consistency *vs.* Pairwise label consistency.** We substitute the triplet-level label consistency term $\psi_{triplet}$ with a pairwise label consistency term $\psi_{pair}$ similar as that in SG-CRF, and this model variant is named as *JM-SGG (pair)*. In specific, following the idea of SG-CRF, this model measures the consistency of each label pair (*i.e.* head object label and tail object label, head object label and relation label, tail object label and relation label) by concatenating two labels' prototypes and mapping the concatenation to a scalar energy $e$ with a nonlinear MLP, and the potential term for the label pair is defined as $\exp(-e)$. This model is compared with the standard JM-SGG model which measures the ternary label consistency of each head-relation-tail triplet. In Tabs. D, E and F, we can observe that the standard JM-SGG outperforms JM-SGG (pair) with a clear margin, which illustrates that modeling ternary label consistency can indeed enhance the expressiveness of JM-SGG and improve its performance.
>
> The results below will be added to the revised draft together with the analysis above.
>
>
>
> Table D: Predicate Classification performance with graph constraint on Visual Genome (ablation study for relation potential function).
>
> |Method|mR@50|mR@100|R@50|R@100|
> |:----:|:----:|:----:|:----:|:----:|
> |JM-SGG (TransE)|22.7|25.4|68.1|68.8|
> |JM-SGG (pair)|23.2|26.1|68.5|69.4|
> |JM-SGG|**24.9**|**28.0**|**70.8**|**71.7**|
>
>
> Table E: Scene Graph Classification performance with graph constraint on Visual Genome (ablation study for relation potential function).
>
> |Method|mR@50|mR@100|R@50|R@100|
> |:----:|:----:|:----:|:----:|:----:|
> |JM-SGG (TransE)|11.5|13.2|41.7|42.2|
> |JM-SGG (pair)|11.9|13.5|42.3|42.8|
> |JM-SGG|**13.1**|**14.7**|**43.4**|**44.2**|
>
>
> Table F: Scene Graph Generation performance with graph constraint on Visual Genome (ablation study for relation potential function).
>
> |Method|mR@50|mR@100|R@50|R@100|
> |:----:|:----:|:----:|:----:|:----:|
> |JM-SGG (TransE)|7.9|9.6|27.6|29.8|
> |JM-SGG (pair)|8.5|10.4|27.9|31.0|
> |JM-SGG|**9.8**|**11.8**|**29.3**|**32.2**|

---

> > ### Author Response · Authors · 2021-08-10
> > **Author Response to Reviewer PwTs (Cont.)**
> >
> > **-Q5:** The results from the model without factor update, *i.e.* JM-SGG (w/o FU), is better than the triplet-only model that does one step of factor update, *i.e.* JM-SGG (triplet). This result needs more explanation.
> >
> > **-A5:** Thanks for this good suggestion. We think the superior performance of JM-SGG (w/o FU) over JM-SGG (triplet) is mainly ascribed to following reasons. (1) **The joint scene graph modeling during learning is effective**: Though JM-SGG (w/o FU) does not perform inference-time joint modeling via factor update, it indeed learns the model with the joint likelihoods of whole scene graphs, which can facilitate the model to extract more precise visual representations and derive better potential functions. However, this model capacity is not included in JM-SGG (triplet). (2) **The factor update on a single triplet is biased**: As you suggested, the factor update on a single triplet tends to update object and relation labels to those label combinations commonly occurred in the training set, *e.g.* <man, riding, bike>, which can mislead the specific case at test time like <man, carrying, bike>. Therefore, the one-step factor update during inference can sometime hurt the performance of JM-SGG (triplet). In summary, these two reasons both enhance the motivation of performing joint scene graph modeling, and we will explicitly state them in the revised version.
> >
> > **-Q6:** The related work section is lacking since it doesn't mention how the proposed method is different and/or better compared to existing works.
> >
> > **-A6:** Thanks for pointing out this drawback. In the revision, we will thoroughly compare our method with existing works and discuss how our method makes improvement. In particular, we will focus on discussing how our model achieves stronger expressiveness than two closely related works, Scene Graph Generation via Conditional Random Fields (SG-CRF) by Cong et al. 2018 and Energy-Based Learning for Scene Graph Generation (EBM-SGG) by Suhail et al. 2021. Specifically, similar to our work, these two works both seek to perform joint scene graph modeling. However, compared to SG-CRF, the proposed JM-SGG additionally models the ternary dependencies within all head-relation-tail triplets, which is achieved by defining the ternary potential function using the knowledge graph embedding technique. This ternary potential term enables JM-SGG to perform relation reasoning and derive more accurate scene graph predictions. Compared to EBM-SGG, JM-SGG additionally models the dependency between each semantic label and its corresponding visual representation using the unary potential function, which enables the more precise inference of each object and relation label based on exact visual grounding.
> >
> > **-Q7:** It will be good to include additional results with ResNext backbone.
> >
> > **-A7:** Thanks for this great suggestion. In a recent work, BGT-Net: Bidirectional GRU Transformer Network for Scene Graph Generation by Dhingra et al. 2021, authors verified the performance improvement when the backbone is switched from VGG16 to ResNext-101. This experiment is important to understand the influence of backbone architecture on scene graph generation model. We are running the experiments, but as the model with ResNext-101 backbone is quite computationally intensive, we have not got the results. We will finish the experiments using ResNext-101 backbone in the next two weeks and add the results to the revised draft.

---

> > > ### Comment · Reviewer_PwTs · 2021-08-31
> > > **Thanks for the response**
> > >
> > > Just wanted to acknowledge that I have read the response and appreciate the additional details.

---

### Official Review · Reviewer_TbVc · 2021-07-15

**Rating:** 6
**Confidence:** 3

**Summary:**

The paper proposes a new method for supervised scene graph generation from images. The new method models dependencies between objects and their relations as a conditional random field (CRF), conditioned on the output of an object detector. To facilitate inference in the GRF, the deterministic prediction by the object detector is used to initialize a mean field approximation to the posterior. This is then updated iteratively via message passing. The method is shown to yield improved performance in relationship retrieval on the Visual Genome dataset.

**Limitations And Societal Impact:**

Limitations and societal impact are adequately addressed.

**Main Review:**


### Method
The intuition behind the proposed method is clear, and the design choices made in its implementation are plausible. The theoretical analysis is correct as far as I can tell. The use of the mean field approximation is a limitation, but one that is reasonable and also discussed in the paper.

### Originality
The introduction and related work section claims that the proposed method is the first to model dependencies across the entire scene graph. I don't believe that this is the case: At the least, Suhail et al. [33] propose an energy based model that uses a graph neural network to assign energy values to scene graphs, which will be able to model arbitrary dependencies between the nodes. They also use a similar inference procedure, by first obtaining an initial scene graph via a deterministic predictor, and then iteratively updating it to maximize the likelihood (energy) under the probabilistic model. While the proposed method may be the first to model dependencies via a traditional graphical model like a CRF, and there may be some benefits to that, such a claim would need to be discussed and supported by evidence.

### Empirical Evaluation
The model is evaluated in the standard setting of relationship retrieval on the Visual Genome dataset, with Table 1 indicating that the proposed methods performs almost universally better than previous methods. However, given that the authors report baseline numbers for VCTree, VCTree-TDE, and VCTree with the EBM loss from [33], I am very confused that they chose to not also report the results for VCTree-TDE with EBM loss [33]. This appears to be the current state of the art on this benchmark, and also beats the proposed method on most of the reported metrics. Am I missing any reason why this wouldn't be a fair comparison?

More generally, the robustness of the evaluation could be improved by including another dataset, such as GQA.


### Clarity
Overall, the proposed method is explained in a clear and reproducible way.
The paper could benefit from another editing pass, e.g., the caption of Fig. 2(c) is broken.
The readability of some of the math, especially formulas (12) and (13) could be improved, e.g., by avoiding nested subscripts such as $y_{o_h}$, and
merging summation signs.

### Summary
Overall, the paper proposes a plausible method, but the discussion of related work and the empirical evaluation raise questions. I am open to increasing my score if these can be addressed during the response phase.

### Update
In their response, the authors have agreed to remove the claim that they are the first to jointly model scene graphs. They have also demonstrated that their contribution is orthogonal to the unbiased prediction techniques employed in VCTree-TDE, and that the two can be combined to achieve even better performance. This addresses my two main concerns. I have therefore raised my score from 5 to 6.

**Time Spent Reviewing:**

7

---

> ### Author Response · Authors · 2021-08-10
> **Author Response to Reviewer TbVc**
>
> Thanks for your insightful comments and golden suggestions! In the revision, we will polish the paper as you suggested, *i.e.* fixing the caption of Fig. 2(c), avoiding nested subscripts in formulas and merging summation signs in formulas. We respond to your questions and concerns as below:
>
> **-Q1:** The connections and differences with a closely related work, Energy-Based Learning for Scene Graph Generation (EBM-SGG), Suhail et al. 2021, are not adequately discussed.
>
> **-A1:** Thanks for pointing out this important related work. As you suggested, both EBM-SGG and the proposed JM-SGG models can capture arbitrary dependency between the object and relation labels in the scene graph.
>
> One major difference between the two methods is on how the potential function over a head-relation-tail triplet is defined. In EBM-SGG, it uses the message function of GNNs to compute a potential score for each head-relation-tail triplet. Although the message functions are general and flexible, they are not specifically designed for relational reasoning, leading to inferior results. Different from EBM-SGG, we follow an existing knowledge graph embedding method, TransR, to define the potential function on each head-relation-tail triplet. TransR has been proven effective in other relational reasoning tasks such as knowledge graph completion, and thus it **brings stronger inductive bias to our approach for relationship modeling**. Therefore, our approach gets superior results in the experiment.
>
> Besides, another main difference is on how the semantic label of each object or relation is modeled. In EBM-SGG, the scene graph with object and relation labels is constructed independently from the image graph composed of each label’s visual grounding, and these two graphs are encoded by two independent GNNs, which fails to model the dependency of each semantic label on its corresponding visual cue. By comparison, JM-SGG uses the unary potential function to model the dependency of each semantic label on its corresponding visual representation, and thus **the inference of each label is performed based on exact visual grounding**.
>
> In summary, compared to EBM-SGG, JM-SGG is designed based on stronger inductive bias and is more expressive on modeling visual groundings. We will supplement these discussions to the revised draft.
>
> **-Q2:** The comparison with VCTree-TDE w/ EBM loss should also be included.
>
> **-A2:** Thanks for your kind reminder. As you suggested, VCTree-TDE w/ EBM loss achieves state-of-the-art performance on the Visual Genome benchmark, and it ought to be included for comparison. However, in the original draft, we exclude the comparison with this strong method mainly because of the following reasons: (1) The VCTree-TDE w/ EBM loss model combines the contributions of two works, the *Total Direct Effect (TDE) for unbiased prediction* by Tang et al. 2020 and the *EBM-based joint scene graph modeling* by Suhail et al. 2021; (2) The proposed JM-SGG model mainly focuses on *CRF-based joint scene graph modeling* and does not perform unbiased prediction, which makes it **fairer to be compared with VCTree-TDE and VCTree w/ EBM loss but not their combination, VCTree-TDE w/ EBM loss**.
>
> Here, we further study another configuration of our JM-SGG model that can lead to fair comparison with VCTree-TDE w/ EBM loss. Specifically, following the idea of TDE, we additionally derive the counterfactual predictions of relation labels by wiping out the observed object representations, and such predictions can be regarded as the dataset bias, *e.g.* the dominant relation types between two object categories. After that, we subtract such biased predictions from the original relationship predictions of JM-SGG model, which derives the unbiased predictions of JM-SGG. We name this new model configuration as **JM-SGG (unbiased)**. From the results in Tabs. A, B and C below, we can observe that JM-SGG (unbiased) outperforms VCTree-TDE w/ EBM loss on 4 of 6 settings for unbiased prediction (*i.e.* the settings using metric mR@$k$). These results verify that when the TDE-based unbiased prediction is added to JM-SGG model, it can perform comparably or even better than the VCTree-TDE w/ EBM loss model on unbiased prediction tasks.
>
> The results below will be added to the revised draft together with the analysis above.
>
>
> Table A: Predicate Classification performance with graph constraint on Visual Genome.
>
> |Method|mR@50|mR@100|R@50|R@100|
> |:----:|:----:|:----:|:----:|:----:|
> |VCTree-TDE w/ EBM loss|26.7|30.0|68.7|70.1|
> |JM-SGG (unbiased)|**27.8**|**30.9**|**72.3**|**73.0**|
>
>
> Table B: Scene Graph Classification performance with graph constraint on Visual Genome.
>
> |Method|mR@50|mR@100|R@50|R@100|
> |:----:|:----:|:----:|:----:|:----:|
> |VCTree-TDE w/ EBM loss|**18.2**|**20.5**|43.1|43.8|
> |JM-SGG (unbiased)|17.6|20.0|**44.1**|**44.9**|
>
>
> Table C: Scene Graph Generation performance with graph constraint on Visual Genome.
>
> |Method|mR@50|mR@100|R@50|R@100|
> |:----:|:----:|:----:|:----:|:----:|
> |VCTree-TDE w/ EBM loss|9.7|11.6|28.4|31.9|
> |JM-SGG (unbiased)|**10.3**|**12.5**|**29.6**|**32.7**|
>
>
>
> **-Q3:** The robustness of the evaluation could be improved by including another dataset, such as GQA.
>
> **-A3:** Thanks for this great suggestion. The GQA dataset can evaluate the model using denser scene graphs with more object categories and relation types, which will definitely provide more robust evaluation. We are running the experiment, but as the dataset is quite large, we have not got the results. We will finish the experiments on GQA in the next two weeks and add the results to the revised draft.

---

> > ### Comment · Reviewer_TbVc · 2021-08-27
> > **Score Increase**
> >
> > Thank you for your detailed response. It addresses my main concerns, and I have therefore increased my score to 6.

---

### Official Review · Reviewer_Bh7i · 2021-07-16

**Rating:** 6
**Confidence:** 4

**Summary:**

This paper proposes a joint prediction of the entire scene graph that fully captures the dependency among different objects and relations using a unified conditional random field.  The proposed model can be summarized as starting from joint modeling of the object and relation component label and visual features. Joint modeling uses CRF to model comprehensive dependency of the object using unnormalized likelihood and partition function. In addition, the potential function used for object and relation components computed affinity through a distance-based learnable prototype. Finally, the knowledge graph embedding techniques for projection of different embedding to the same embedding space (e.g. context space to relation space, object space to relation space) have been used.

All the learnable parameters are trained with maximum log-likelihood function and the partition function uses a mean-field variational inference for efficient sampling. Mean-field inference initialized with independent relation labels for triplet component factor update.

Authors have conducted extensive experiments on the Visual Genome dataset and a brief ablation study that provide a good insight into the method.

**Limitations And Societal Impact:**

There are a few limitations of the paper :
1. Authors should conduct more experiments on relatable datasets like GQA or Open Image.
2. Ablation focuses on the joint modeling, factor update, initialization, etc, where the paper assumes  CRM and prototypical embedding helped the network. Ablation should provide the impact of these components.


**Main Review:**

The main idea of the paper for joint modeling of object and relation using CRF and mean-field variance algorithm is novel and significant in the SGG task. The paper is well written, although the introduction and description of the long list of components hindrance its readability. The combination of CRF and knowledge graph technique will lead to exploration of commonality between scene and knowledge graph, thus making it a relevant work in this area.



**Time Spent Reviewing:**

4

---

> ### Author Response · Authors · 2021-08-10
> **Author Response to Reviewer Bh7i**
>
> Appreciate for your insightful comments and great suggestions! We respond to your questions as below:
>
> **-Q1:** More experiments on relatable datasets like GQA and Open Images should be conducted.
>
> **-A1:** Thanks for this good suggestion. In the current draft, we only include the experimental results on Visual Genome dataset, which is a standard and commonly-used benchmark dataset for evaluating scene graph generation models. Indeed, it is better to include more datasets for broader evaluation. For example, the GQA dataset can evaluate the model using denser scene graphs with more object categories and relation types, and the Open Images dataset provides a larger-scale dataset with rich relationship annotations for evaluation. We are running the experiments, but as the datasets are quite large, we have not got the results. We will finish the experiments on GQA and Open Images in the next two weeks and add the results to the revised draft.
>
> **-Q2:** The ablation studies for the CRM-like joint distribution modeling and the prototypical embedding are lacked.
>
> **-A2:** Thanks for pointing out these important ablation studies. As you suggested, the proposed JM-SGG model performs joint modeling of visual representations and semantic labels in a way like the Continuous-space Relevance Model (CRM). In specific, they both seek to model the joint likelihood of observing a set of image regions together with the set of textual labels. To verify the effectiveness of such joint modeling scheme, we further study an independent modeling scheme, in which the likelihood of each image region together with its corresponding object/relation label is modeled independently from other regions and labels, named as **Independent Modeling for Scene Graph Generation** (*IM-SGG*). In the implementation, this model only uses the unary potential function of each object and relation, and it does not perform iterative label updates due to the lack of higher-order potential functions. For fair comparison, we compare IM-SGG with the JM-SGG (w/o FU) configuration (FU denotes factor update), such that the only difference between two models is whether joint modeling is applied or not. From the results in Tabs. A, B and C below, we can observe an obvious performance gain by JM-SGG (w/o FU), which demonstrates the benefits of modeling various image regions and semantic labels jointly.
>
> In order to verify the effectiveness of prototypical embedding, we study another model which independently predicts each object/relation label using a two-layer nonlinear classifier instead of prototypes, named as **IM-SGG (classifier)**. In Tabs. A, B and C, we can observe that the performance of IM-SGG and IM-SGG (classifier) are comparable with each other. These results illustrate that prototypical embedding and neural network classifier are equally effective in our scene graph generation model, while prototype embedding can further enable joint scene graph modeling, *i.e.* towards the JM-SGG (w/o FU) configuration, which obviously enhances the model's performance.
>
> The ablation results below will be added to the revised draft together with the analysis above.
>
>
> Table A: Predicate Classification performance with graph constraint on Visual Genome.
>
> |Method|mR@50|mR@100|R@50|R@100|
> |:----:|:----:|:----:|:----:|:----:|
> |IM-SGG (classifier)|13.3|14.9|62.7|64.4|
> |IM-SGG|13.4|15.2|63.2|65.5|
> |JM-SGG (w/o FU)|**23.7**|**26.5**|**69.1**|**70.2**|
>
>
> Table B: Scene Graph Classification performance with graph constraint on Visual Genome.
>
> |Method|mR@50|mR@100|R@50|R@100|
> |:----:|:----:|:----:|:----:|:----:|
> |IM-SGG (classifier)|7.6|8.1|36.9|37.4|
> |IM-SGG|7.4|7.8|36.3|37.0|
> |JM-SGG (w/o FU)|**12.1**|**13.6**|**42.7**|**43.2**|
>
>
> Table C: Scene Graph Generation performance with graph constraint on Visual Genome.
>
> |Method|mR@50|mR@100|R@50|R@100|
> |:----:|:----:|:----:|:----:|:----:|
> |IM-SGG (classifier)|5.5|6.4|25.4|27.5|
> |IM-SGG|5.2|5.9|24.8|27.1|
> |JM-SGG (w/o FU)|**8.7**|**10.5**|**28.2**|**31.3**|

---

### Official Review · Reviewer_szaq · 2021-07-19

**Rating:** 7
**Confidence:** 4

**Summary:**

This paper presents a conditional random field (CRF) based joint modeling of objects and relations for the task of scene graph generation. Unlike previous relation discovery methods, this paper employs embedding-based relation feature representation, and thus enables an unified modeling of the unitary and clique potential functions. Using MCMC sampler and mean-field variational inference, the proposed JM-SGG can effectively update the relation triplets utilizing the contextual label dependency. The experiments show that JM-SGG has good performances on the relationship retrieval task, even under the zero-shot setting.

**Limitations And Societal Impact:**

Yes, the authors have discussed the limitations and potential negative societal impact of this work in the section of the conclusion, and in the supplementary material.

**Main Review:**

Overall, this study is helpful to researchers in the fields of visual understanding, especially scene graph generation. It is happy to see how the probabilistic graph model is successfully integrated into scene graph generation. This paper is well written, clearly illustrated, and appropriately structured. The experimental results also validate the claims.

But in my opinion, conditioned random field and mean-field variational inference are not at the first time applied into scene graph generation. For example, Cong et al. 2018, Scene Graph Generation via Conditional Random Fields has introduced some important techniques used in this paper. Even though this paper was just `published' in arxiv, it would be appreciated to discuss their technical connections, and how the proposed method outperforms.

Another concern lies in the experiments. At first, why not in addition report the results by non-graph constraint? Each ordered object pair should contain more than one relation if the relation label cannot convincingly exclude each other. For example, <person, has, jean> is not wrong, even though <person, wearing, jean> may be a more precise description. It would be nice if the proposed JM-SGG can capture the most possible relations for a particular object pair.

**Time Spent Reviewing:**

3 hours

---

> ### Author Response · Authors · 2021-08-10
> **Author Response to Reviewer szaq**
>
> Thanks for your valuable comments and golden suggestions! We respond to your questions as below:
>
> **-Q1:** The connections and differences with a highly related work, Scene Graph Generation via Conditional Random Fields (SG-CRF), Cong et al. 2018, are not adequately discussed.
>
> **-A1:** Thanks for pointing out this highly related literature. Both SG-CRF and the proposed JM-SGG model utilize conditional random fields to model the joint distribution of entire scene graphs, and they both use a mean-field algorithm to iteratively refine scene graph predictions. However, SG-CRF only models unary and pairwise label likelihoods in the *Semantic Compatibility Network*. In scene graph generation, what we care about is each head-relation-tail triplet, and hence an expressive model is expected to have a ternary term modeling the connection of each triplet. Inspired by knowledge graph embedding methods, **we introduce such a ternary term on each head-relation-tail triplet**, allowing our approach to perform relational reasoning and achieve superior results on various scene graph generation tasks. We will supplement these discussions to the revised draft.
>
> **-Q2:** The w/o graph constraint performance of JM-SGG model is required to better evaluate whether it can capture the most possible relations between a particular object pair.
>
> **-A2:** Thanks for this good suggestion. We provide the w/o graph constraint performance of various methods in the following tables. The performance of IMP+, FREQ, Motifs, KERN and GB-Net-$\beta$ are taken from the paper of GB-Net. In the following tables, we can observe that the proposed JM-SGG model outperforms existing methods with a clear margin, which illustrates that JM-SGG can well capture the most possible relations between a particular object pair.
>
> The results below will be added to the revised draft together with the analysis above.
>
>
> Table A: Predicate Classification performance without graph constraint on Visual Genome.
>
> |Method|mR@50|mR@100|R@50|R@100|
> |:----:|:----:|:----:|:----:|:----:|
> |IMP+|20.3|28.9|75.2|83.6|
> |FREQ|24.8|37.3|71.3|81.2|
> |Motifs|27.5|37.9|81.1|88.3|
> |KERN|36.3|49.0|81.9|88.9|
> |GB-Net-$\beta$|44.5|58.7|83.5|90.3|
> |JM-SGG|**46.2**|**60.9**|**84.8**|**92.1**|
>
>
> Table B: Scene Graph Classification performance without graph constraint on Visual Genome.
>
> |Method|mR@50|mR@100|R@50|R@100|
> |:----:|:----:|:----:|:----:|:----:|
> |IMP+|12.1|16.9|43.4|47.2|
> |FREQ|13.5|19.6|40.5|43.7|
> |Motifs|15.4|20.6|44.5|47.7|
> |KERN|19.8|26.2|45.9|49.0|
> |GB-Net-$\beta$|25.6|32.1|46.9|50.3|
> |JM-SGG|**26.5**|**33.4**|**48.7**|**52.3**|
>
>
> Table C: Scene Graph Generation performance without graph constraint on Visual Genome.
>
> |Method|mR@50|mR@100|R@50|R@100|
> |:----:|:----:|:----:|:----:|:----:|
> |IMP+|5.4|8.0|22.0|27.4|
> |FREQ|5.9|8.9|25.3|30.9|
> |Motifs|9.3|12.9|30.5|35.8|
> |KERN|11.7|16.0|30.9|35.8|
> |GB-Net-$\beta$|11.7|16.6|29.3|35.0|
> |JM-SGG|**13.3**|**18.5**|**32.0**|**37.4**|

---

### Decision · Program_Chairs · 2021-09-27

**Decision:**

Accept (Poster)

**Comment:**

Thank you for submitting your work to NeurIPS. The paper introduces a supervised approach to scene graph generation from images: objects and relations among them are captured using a CRF conditioned on a (deep) object detector and using e.g. distance-based learnable prototype. Overall, the rolling discussion helped to clarify many of issues raised by the reviewers. This is a solid but also somewhat incremental paper that, and this is a big advantage of the paper, shows the value of hybrid methods for challenging AI tasks. In any case, please incorporate your feedback from the rolling discussion into the final version.